# Teaching programming using eduScrum methodology

Patrik Voštinár

Department of Computer Science, Matej Bel University, Banska Bystrica, Slovak Republic,
Slovak Republic

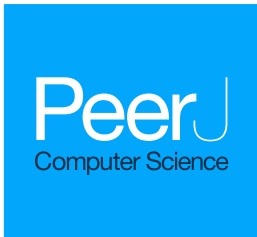

## ABSTRACT

There are a large number of professions in the world today. Some professions are disappearing, and some new ones are emerging. However, they all have something in common: the need to manage them. Throughout its history, humanity has developed several constantly changing forms of management. For this reason, school absolvents must enter the labour market with skills already sufficiently developed, such as communication, cooperation, teamwork, responsibility, and the ability to plan their work. The article focuses on the issue of teaching programming through mobile applications and basic robotics through the innovative form of teaching-EduScrum. The EduScrum methodology is based on the agile software development method Scrum, which develops soft skills. The article describes our experience with this teaching in computer science classes. We established several hypotheses evaluated using descriptive statistics on a sample of 251 students. The main objective of the research is to verify whether teaching computer science in primary and secondary schools using the eduScrum methodology is more suitable than the classical-frontal teaching of computer science. The research showed that secondary school students preferred the eduScrum methodology more than traditional frontal teaching and the primary school students preferred traditional frontal teaching.

## INTRODUCTION

Information and communication technologies (ICT) are already a standard part of all levels of education. Every school has at least one computer room, and different subjects have special ICT tools to make teaching more visual. Computer science teachers are increasingly using a variety of motivational teaching aids, such as robotic kits, mobile devices, and various didactic software, to motivate students to code. There are several robotic kits and didactic software types on the market today.

According to the International Standard Classification of Education (ISCED), teaching programming in regional education currently belongs to the thematic unit of algorithmic problem-solving. In primary education at the first level of primary school, students should already be able to think about algorithms, find algorithmic solutions to problems, and create instructions and programs according to given rules. They can use various visual programming environments, *e.g.*, Scratch Junior, Baltik, Robot Emil, and other robotic aids, such as Bee-Bot, Blue-Bot, Probot, and mBot Tiny.

Corresponding author
Patrik Voštinár,
patrik.vostinar@umb.sk

Computer science teachers have a wide range of software and educational tools available to teach programming at the secondary level. These websites and tools are starting to be sought after by computer science teachers and teachers of other subjects.

Thanks to various projects and foundations, schools have more and more robotic tools and teaching materials that they can use to enrich the teaching of programming. A national IT Academy project in Slovakia has been running since 2016, involving five universities and many primary and secondary schools. Selected schools received free innovative equipment, including Probot robots, Lego EV3, micro:bit, Picoboard, Raspberry Pi, Arduino, and other educational aids, not limited to computer science. In the Czech Republic, a similar project under the Operational Programme Research, Development, and Education entitled Support for the Development of Computational Thinking is underway. More about the activities and outputs of this project can be found in *Blaho (2019*, *2021*), *Nagyová (2021)*, and *Vaníček (2021)*. Every year, teachers from primary and secondary schools can receive funding to buy educational tools or pay for licenses through the Meet and Code initiative.

Unfortunately, motivational aids such as educational robots and kits are not enough to motivate some students to study programming. There is a worldwide need for more IT professionals. Even if they manage to find candidates for computer scientist positions, these graduates often need more ability to communicate, collaborate, critically evaluate their shortcomings, and have better accountability. For this reason, the eduScrum methodology for coaching learners has been developed based on the agile Scrum software development method, which is increasingly preferred in the IT sector. Teaching using the eduScrum methodology is simple because it sets out WHAT rather than HOW the students will learn. Teaching promotes the so-called soft skills that are often lacking in computer science graduates. In the academic environment, various agile methods have been tested in recent years in the teaching process. *McAvoy & Sammon (2005)* and *Sharp & Lang (2018)* have described positive teaching experiences with agile methods. For example, Scrum methodologies have been discussed by *Rodríguez, Soria & Campo (2016)* and *Von Wangenheim, Savi & Borgatto (2013)*. Scrum method education using Lego simulation has been addressed by *Bourdeau, Romero-Torres & Petit (2021)* and *Krivitsky (2017)*, who showed an interesting approach to explaining Scrum methodology. *Ternikov*'s *(2022)* proposed methodology reveals explicit information about the combinations of "soft" and "hard" skills required for different professional groups. These findings provide valuable insights for supporting educational organizations, human resource (HR) specialists, and state labour authorities in renewing existing knowledge about IT professionals' skill sets. In education, a modified eduScrum project management method is currently used based on the most popular agile method, Scrum. The described researches showed that the use of agile methods in teaching abroad is more interesting, more suitable for students as they acquire better soft skills than in classical conventional teaching.

## TEACHING MOBILE APPLICATION PROGRAMMING

Mobile phones and tablets are hugely popular with people of all ages. It is common to observe young children as young as three years old using mobile phones to watch videos

intended for children. Furthermore, it has become a norm for primary school children to possess mobile phones that their parents can utilize to get in touch with them or locate them using GPS technology.

Using mobile devices in the teaching process naturally increases the motivation of the students to learn programming. At present, multiple development environments are available to create mobile applications. We have previously discussed the pros and cons of mobile learning in our publications including *Voštinár & Hanzel (2017)* and *Voštinár (2017)*.

Currently, native application development (Java/Kotlin for Android, Swift/Objective-C for iOS) is preferred before hybrid application development using, for example, React Native, Xamarin, PhoneGap, Titanium, Rhodes, Native Script, Progressive web application, and Flutter. Each of the environments and languages mentioned above has its advantages and disadvantages. We recommend App Inventor or the official Android Studio environment to learn mobile application development for the Android operating system. App Inventor is a free online programming environment for creating apps for the Android operating system. To create apps, sign in with a Google account (*e.g.*, Gmail account) at http://appinventor.mit.edu. App Inventor uses block programming. Therefore, it is very similar to the Scratch programming environment students are exposed to in primary school. Figure 1 shows the App Inventor environment. The App Inventor consists of Designer and Blocks parts. Designer consists of a tool Palette (list of available components), Viewer (how it will look on the phone), Components (a list of all used components), Properties (the place where we set the properties of each component).

Several researches and studies have addressed using this environment in the teaching process. *Wober (2014)* wrote about the positive results of the pilot program in his research. *Gray et al. (2012)* wrote about the experience of teaching in the App Inventor environment in their publication. *Georgiev (2019)* from Ruse University researched the use of the MIT App Inventor programming environment at different levels of education. *Kim & Lee (2017)* studied the possibility of incorporating App Inventor into the national curriculum for elementary school students in South Korea.

The Android Studio development environment has been the recommended development environment for the Android operating system since 2013. Applications can be created in Java or Kotlin. We can test the created applications directly on mobile devices or use the emulator of Android devices. Studies dealing with teaching mobile application programming using Java or Kotlin have been done, *e.g.*, by *Barbosa et al. (2007)*, *Tillmann et al. (2012)*, *Al-Khalifa, Faisal & Al-Gumaei (2019)*, and others.

## ROBOTIC AIDS

Using various robotic kits in teaching has become increasingly attractive in recent years. It is common if these devices are already used in pre-schools or the first stage of primary school. *Stoffová & Zboran (2018)*, *Bezáková, Hrušecká & Hrušecký (2021)*, *Merkouris & Chorianopoulos (2018)*, and *Klimeková et al. (2018)* have written about the appropriateness of using robotic devices in primary school.

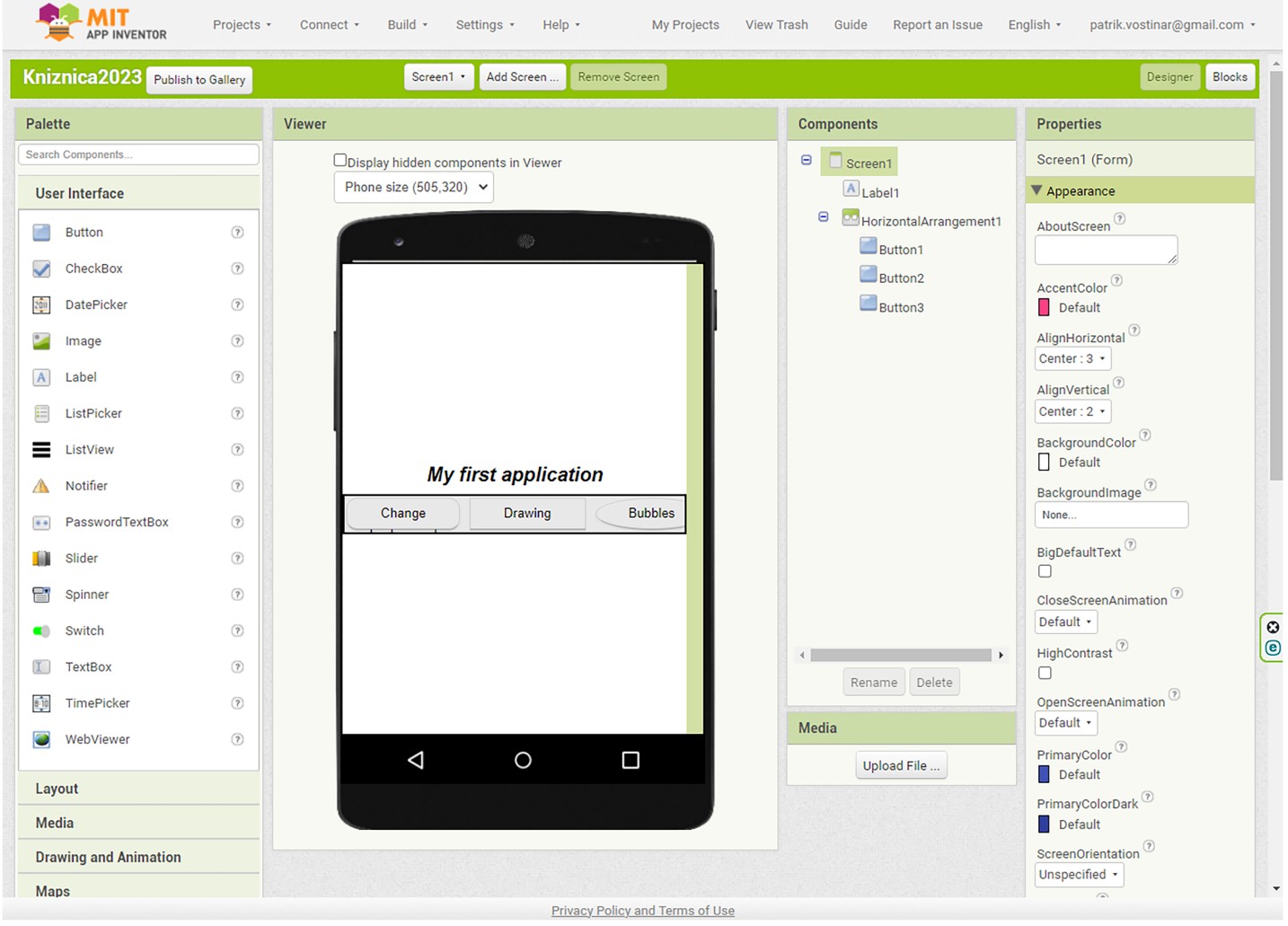

**Figure 1 App Inventor programming environment.**

Currently, there are a large number of programmable robots, *e.g.*, LEGO Mindstorms kits, Bee-Bot, ProBot, Sphero, Ozobot, robots created by Makeblock (Codebot, Airblock, mBot), and Phiro Pro. Each of these robots has its advantages and disadvantages. Figure 2 shows robots Ozobot (top left), mBot (top centre-blue robot), Lego EV3 (right), Sphero (bottom left-ball) and Edison (centre bottom-orage).

The NXT EV3 robots from Lego are among the world's most famous robotic aids. In June 2021, the sale of the most popular Lego Mindstorms EV3 kit ended and was replaced by Lego Mindstorms 51515, which supports the Scratch block-based programming environment. Using Lego building blocks in education helps students acquire manual and technical skills and theoretical knowledge. Kits can help teach various subjects such as physics, chemistry, and computer science. *Korkmaz (2018)*, *Zhan & Hsiao (2020)*, and *Kunduracıoğlu (2018)* also confirmed the suitability of using Lego building blocks in the teaching process in their studies.

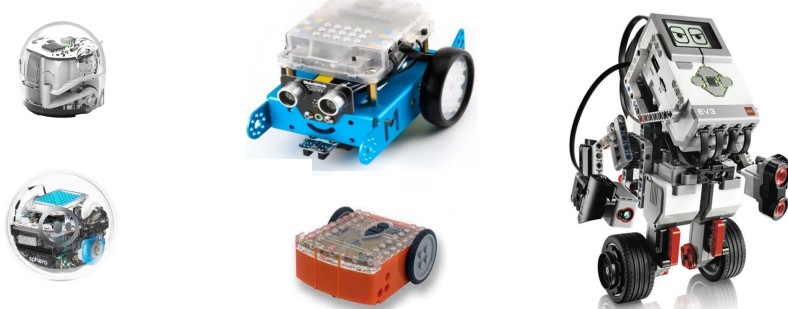

**Figure 2  Most used robots for teaching in Slovak Republic.**

Other popular robotic devices are from MakeBlock. The main advantage of these devices is their low price compared to other robots and the possibility of programming them in the mBlock software or using an app on a mobile device (iOS and Android). The company's best-selling robot kits are mBot and mBot Ranger, which are based on Arduino microcontroller hardware and can be programmed in code blocks and development environments supporting Arduino microcontroller programming. *Pisarov & Mester (2019)*, *Numanoğlu & Keser (2017)*, and *Sáez-López, Sevillano-García & Vazquez-Cano (2019)* researched the positive experiences of using mBot robots.

Ozobot is a programmable microrobot that develops creative and logical thinking and helps students learn to code. Like Lego Mindstorms and MakeBlock robots, it is hugely popular with students around the world, and its main advantage over its competitors has been the ability to program using colour-coded markers. Ozobot has won several awards for best robotic toy. The implementation of Ozobot robots in education in Slovak and Czech schools has been reported by *Picka, Dosedla & Stuchlikova (2020)*, *Žáček & Smolka (2019)*.

Other widespread kits designed for the second grade of elementary and high schools include Sphero, Edison, and Phiro Pro robotic kits. Table 1 compares robots suitable for teaching in primary and secondary schools.

## TEACHING PROGRAMMING THROUGH EDUSCRUM

The eduScrum methodology was developed in 2013 by physics and chemistry teachers in the Netherlands. The eduScrum methodology is based on the agile software development method Scrum (*Sutherland & Schwaber, 2007*). Scrum is one of the most widely used methods for planning and managing projects requiring agile development. Scrum has defined techniques, processes, and artefacts that we must follow to manage a project using this method.

EduScrum is a methodology for guiding students in which the teacher delegates responsibility for the learning process to the students. Students in the class are divided into groups of four and work collaboratively on studying new material, working on exercises, solving problems, and presenting their findings to the class. The teacher provides instructions to the students but does not give specific details on completing the task. The

**Table 1 Comparison of robotic aids.**

| | Age | Software | Programming languages | Sensors | Price |
|---|---|---|---|---|---|
| **Lego EV3** | 10+ | EV3 Lab, MakeCode Mindstorms, mBlock, Scratch | Block based, microPython | Light, color, ultrasonic, touch and IR sensors, gyroscope | 400–900€ |
| **mBot** | 10+ | mBlockly, mBlock | Block based, Python, Arduino | Motion, temperature and ultrasonic sensors, line tracking, Bluetooth/WiFi | 90–140€ |
| **Ozobot evo** | 14+ | Color codes, Ozobot, OzoGroove, OzoBlockly | Block based, ozocodes, JavaScript | Line, ultrasonic, color, proximity, optical, RGB LED, Bluetooth | 100–120€ |
| **Sphero bolt** | 14+ | Sphero edu | Block based, JavaScript | Gyroscope, light and IR sensors, Bluetooth, LED display, compass | 99–220€ |
| **Edison** | 4+ | EdWare | Barcodes, EdBlocks, EdPy, EdScratch | Light, sound and IR sensors, line tracking, obstacle avoidance | 35–40€ |

students decide for themselves what to do in class and what to do for homework. The primary importance of eduScrum teaching is strengthening soft skills-communication, collaboration, presentation, and self-reflection. EduScrum clearly outlines the project status, ensuring each team member and teacher knows each other's tasks.

Most schools in Slovakia teach computer science and other subjects through frontal teaching. In Slovakia, the innovative teaching method was introduced in Prešov in 2015 and later at the Technical University in Košice. The goal was to encourage students to acquire soft skills and motivate them to study computer science.

## EduScrum roles

The students are divided into teams of four (the most ideal number). Each team works towards a set goal. Team members are jointly responsible for meeting the acceptance criteria, with no one telling them how to implement the goals (they are self-organized). In each team, there is one student elected to the position of eduScrum Master (can be elected by the teacher or also by the students–*e.g.*, by voting), who has the following task:

- Keep an up-to-date task board-Flipboard and keep it available so that it does not get lost,
- support and motivate the team,
- collaborate with the teacher (in eduScrum, the teacher has a position called Product Owner).

The teacher is responsible for setting learning objectives, monitoring progress, and assessing students. His position is called the same as in the Scrum methodology-Product Owner. He provides the teams with recommendations for learning materials (however, the learners themselves decide whether to use them or find other, more suitable materials for them), answers questions (tries to guide the learners to a solution, *e.g.*, with an example, not by providing a specific solution).

## EduScrum events and activities

Teaching using the eduScrum methodology is divided into time-bound events with a clearly defined goal. None of these events should be missed. Teaching is divided into

regular cycles called Sprints (lasting approximately one or two months, determined by the teacher). Each Sprint event consists of:

- Sprint planning–The Product Owner divides students into teams (or changes teams) and introduces the learning objective and assessment conditions. The team members divide the tasks (decomposition of the problem into smaller parts). They record the tasks on the Flipboard.
- Stand Up–activity at the beginning of the lesson, which should last at most 1–1.5 min per student. Each member updates the Flipboard in this activity and answers the following questions as a team: '*What did I do last hour to meet my Sprint goals? What am I going to do in this class? Did I have a problem with something?*'
- Sprint Review–Teams present their Sprint solution to the class (in 5 min). The Product Owner will then evaluate whether they have met the objectives of the Sprint event. Each team member must say what they did.
- After completing the Sprint, the students are evaluated to assess their performance. This activity is known as Retrospective. The students engage in team discussions, responding to the following questions: '*What was good? What could have been done better? What will we improve in the next Sprint event?*' The Product Owner should walk between teams and constantly check that students answer the above questions.

In addition to the events in the eduScrum methodology, students use a variety of artefacts:

- Product Backlog–is a document listing learning objectives and working methods. Figure 3 shows first Sprint backlog.
- Flipboard-"Division of tasks" refers to dividing and assigning different tasks to individuals or groups. There can be a article or electronic version. Team members utilize it to stay informed of their classmate's current tasks. Using this artefact, the product owner can judge whether a given team will be able to meet the goals by the end of the Sprint event. Figure 4 shows a Flipboard.
- The definition of completeness-is located in the Flipboard; the team members defined it during the Sprint planning event. It assesses when the work is completed relative to the current learning objective in a specific Sprint event.

## METHODOLOGY AND RESEARCH

We started the first teaching using the eduScrum method in the academic year 2018/2019 in four secondary schools and one primary school. According to the agreement with the school's founder, the teaching was mainly focused on creating mobile applications for the Android operating system. Kotlin was used as the programming language. After an agreement with the primary school management and the project leader, we started using the free online environment App Inventor at the primary school. In the academic year 2020/2021, we modified the programming teaching content in some schools. We have

# Sprint 1 backlog

The Scrum Master is responsible for the running of the whole group. Each member of the group is responsible for the progress of the whole group. Don't forget the stand-ups.

## What you will learn

App Inventor development environment

Create a simple mobile app and test it on your mobile phone/tablet

Scratch block language

Define and use variables, text strings, numbers in a program.

## Goal, what needs to be achieved.

Based on this content, create small tasks in your team and record them on the board. For each lesson, each member of the group will have one task to complete for that lesson. When creating tasks, remember to be creative, don't limit yourself.

Working with GUI (Graphical User Interface) in App Inventor.

Working with numbers and text labels.

Working with variables.

## Acceptance criteria from the teacher's perspective

As a teacher, what I need from everyone at the end of the sprit is this:

Presenting a working demo example and everyone briefly tells what they have learned and done in this sprint. The presentation will be in front of the whole class, the teacher or principal, other teachers, and parents.

## Guides and aids

You can find tutorials for various (not all below) apps and games at www.medium.com (give a keyword search for App Inventor Tutorial Harpuna – tutorial special for eduScrum teaching of App Inventor).

There are even more tutorials in English here:

http://appinventor.mit.edu/explore/ai2/tutorials.html.

**Figure 3** **Sprint backlog.**                               

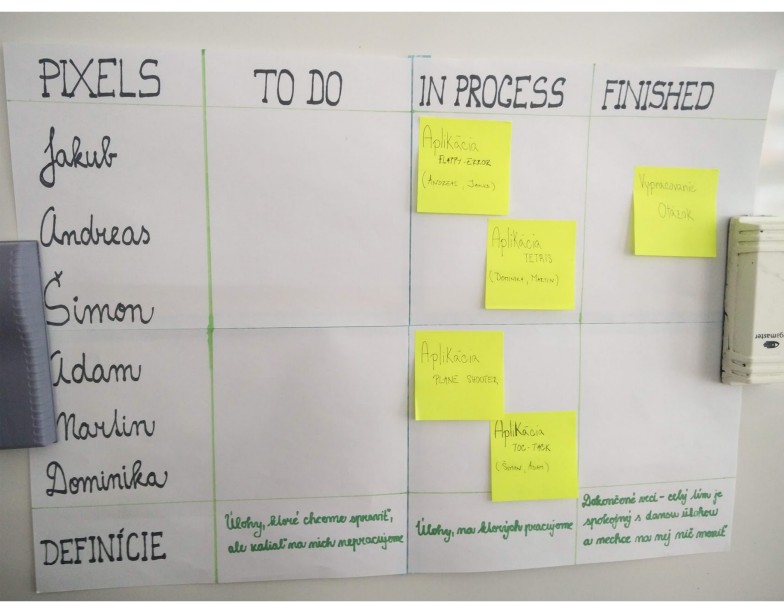

**Figure 4** **Flipboard of the teams.**                     

started teaching robotics programming (micro:bit, Lego EV3, Ozobot, mBot) instead of mobile apps. In all but one school, the students were divided into groups of four, so we randomly selected students for the position of Scrum Master, who then chose the team members. At one secondary school in Zvolen, there were 5-6 member groups, mainly due to the large number of students in the class (31 students).

The research aims to determine if teaching computer science in primary and secondary schools using eduScrum is more effective than classical-frontal teaching.

After completing the course, primary and secondary school students were given an anonymous questionnaire to fill out using Google Forms. The questionnaire contained 22 questions and was completed by 198 out of 251 students taught using the eduScrum method. The research was carried out in seven different schools for 3 years. In secondary schools, the teaching was carried out for two hours per week for a half year, and in primary schools for one hour per week for the whole academic year.

A questionnaire of our design was used to obtain empirical data, which were processed in Microsoft Excel using descriptive statistics methods. In evaluating the responses to the chosen items, we utilized the Mann-Whitney U test and the Chi-Square test.

In our research, we set several hypotheses, which we investigated, *e.g.*, the differences between the answers of 106 boys and 92 girls and the differences between the answers of 68 students from primary schools and 130 students from secondary schools.

The results of the questionnaire and a description of the statistics are available at https://github.com/alabama11/eduscrum-research/wiki.

Using the questionnaire item, '*Compare regular teaching and eduScrum-which do you prefer?*', we investigated the difference between the answers of boys and girls, students from primary and secondary schools. Students could choose whether they like the traditional form or eduscrum more. They had five options to choose from. Table 2 shows the hypotheses for this questionnaire item.

Table 3 shows the hypotheses for the questionnaire item: '*Did you frequently require assistance from your teacher?*' Students could choose from four options (yes, rather yes, rather no, no). Table 4 shows the hypotheses for the questionnaire item '*How did you like App Inventor?*' Students could answer this question on a Likert scale (did not like at all-super). Table 5 shows the hypotheses for the questionnaire item '*Would you like to continue with App Inventor?*' Students could choose from five options (yes, rather yes, rather no, no, don't know/didn't do).

When comparing the responses of primary and secondary school students, we rejected the null hypothesis for several items: '*Compare regular teaching and eduScrum-which do you prefer?*, *Did you often need the teacher's help?*, *How did you like App Inventor?*, *Would you like to continue with App Inventor?*' For all of them, we took the arithmetic mean and found that secondary school students compared to primary school students:

- prefer the eduScrum methodology to conventional teaching,
- more students needed the help of a teacher,
- They liked the App Inventor environment more,
- They think they have gotten better at programming,

**Table 2 Hypotheses for the questionnaire item** *Compare regular teaching and eduScrum-which do you prefer?*

| $H_0$ | $H_1$ | Probability of error | The null hypothesis |
|---|---|---|---|
| The comparison of frontal teaching and eduScrum methodology is similar for boys and girls. | The comparison of frontal teaching and eduScrum methodology is significantly different for boys and girls. | 10.18% | Accept |
| A comparison of frontal teaching and the eduScrum methodology does not differ significantly for primary and secondary school students. | The comparison of frontal teaching and eduScrum significantly differs for primary and secondary school students. | 1.58% | Reject |

**Table 3 Hypotheses for the questionnaire item** *Did you frequently require assistance from your teacher?*

| $H_0$ | $H_1$ | Probability of error | The null hypothesis |
|---|---|---|---|
| The differences in the answers of boys and girls to whether they needed the teacher's help are not significantly different. | There are differences in the responses of boys and girls to whether they needed help from the teacher. | 6.18% | Accept |
| The differences in the responses of primary and secondary school students, whether they needed help from the teacher, are not significantly different. | There are differences in the responses of primary and secondary school students to whether they needed the teacher's help. | 3.38% | Reject |

**Table 4 Hypotheses for the questionnaire item** *How did you like App Inventor?*

| $H_0$ | $H_1$ | Probability of error | The null hypothesis |
|---|---|---|---|
| There are no statistically significant differences between the opinions of boys and girls, whether they liked the App Inventor environment. | There are differences between boys and girls on whether they liked the App Inventor environment. | 89.74% | Accept |
| There are no statistically significant differences between the opinions of primary and secondary school students regarding whether they liked the App Inventor environment. | There are differences between the opinions of primary and secondary school students on whether they liked the App Inventor environment. | 0.00% | Reject |

**Table 5 Hypotheses for the questionnaire item** *Would you like to continue with App Inventor?*

| $H_0$ | $H_1$ | Probability of error | The null hypothesis |
|---|---|---|---|
| There are no significant differences in the responses of boys and girls as to whether they would like to continue learning mobile app programming in App Inventor. | There are differences in the responses of boys and girls as to whether they would like to continue learning mobile app programming in App Inventor. | 27.51% | Accept |
| There are no significant differences in the responses of primary and secondary school students as to whether they would like to continue learning mobile app programming in App Inventor. | There are significant differences in primary and secondary school students' responses regarding whether they would like to continue learning mobile app programming in App Inventor. | 1.74% | Reject |

- would like to continue programming mobile applications in the App Inventor environment.

Table 6 shows the hypotheses for the questionnaire item: *How did you get better at programming?* Students could answer this question on a Likert scale ('I haven't improved my programming at all-I've improved quite a lot').

**Table 6 Hypotheses for the questionnaire item *How did you get better at programming?***

| $H_0$ | $H_1$ | Probability of error | The null hypothesis |
|---|---|---|---|
| Among boys' and girls' responses to the question "How have they improved in programming?" there are no statistically significant differences. | Among boys' and girls' responses to the question "How have they improved in programming?" there are significant differences. | 0.94% | Reject |
| Among the responses of primary and secondary school students to the question "How have they improved in programming?" there are no statistically significant differences. | Among the responses of primary and secondary school students to the question "How have they improved in programming?" there are significant differences. | 0.03% | Reject |

In the questionnaire, we asked students from primary and secondary schools, as well as boys and girls, the following questions to analyze response differences:

- Are you good to work with? Are you a team player?
- Can you be more creative in an eduScrum class than a regular one?
- Is planning important in eduScrum?
- Is task planning important? (We do not mean just eduScrum but in general.)
- The main idea of eduScrum is breaking down a big problem into smaller ones and solving them individually. Do you agree with that?
- Would you like to continue learning with the eduScrum methodology?

We also set null and alternative hypotheses for these questions. For these questions, we accepted the null hypotheses (we also used a Chi-squared test). We did not observe differences between the responses of boys and girls, primary and secondary school students. Based on the responses, it can be inferred that the eduScrum methodology was more enjoyable for most students than traditional computer science classes. In the items focussing on the eduScrum methodology, they answered that this methodology is creative and that it is essential to divide a bigger problem into smaller parts (decomposition), and they also expressed themselves positively in the item of planning in the eduScrum methodology (that planning is essential). The students responded that they thought they were good to work with. Almost all would like to continue developing applications in the App Inventor environment and the eduScrum teaching methodology.

The answers did not surprise us, as the students had already expressed positive views on this teaching methodology during the eduScrum Sprint events, and they had gradually learned to collaborate better and were even more creative in the last presentations than at the beginning.

Table 7 shows the hypotheses for the questionnaire item: *How much did you like eduScrum?*. In this question we used the Mann-Whitney U test to detect differences between the responses of boys and girls, primary and secondary school students, and proposed hypotheses:

Based on the mean of the ratings, boys liked the eduScrum methodology more than girls. Based on the average of the evaluations, students at secondary schools like the

**Table 7 Hypotheses for the questionnaire item *How much did you like eduScrum?***

| $H_0$ | $H_1$ | Probability of error | Value of the test criterion u | The null hypothesis |
|---|---|---|---|---|
| There is no significant difference between boys' and girls' responses in the eduScrum assessment. However, there is a significant difference between boys' and girls' responses in evaluating the eduScrum methodology. | There is no significant difference between the responses of primary and secondary school students in the eduScrum assessment. | 1.96 | 3.566 | Reject |
| There is no significant difference between the responses of primary and secondary school students in the eduScrum assessment. | There is a significant difference between the responses of primary and secondary school students in evaluating the eduScrum methodology. | 1.96 | 2.728 | Reject |

eduScrum methodology more than the regular lesson compared to primary school students. Overall, however, the students liked the eduScrum methodology very much and would like to continue with it.

In the questionnaire item '*How would you rate your team and collaboration?*', we also used the Mann-Whitney U test to detect differences between the responses of boys and girls, primary and secondary school students, and proposed hypotheses in Table 8.

The responses to this questionnaire item were similar to the students' responses on whether they thought they worked well together (questionnaire item: '*Do you think you are good to work with? Are you a team player?*'). In the eduScrum methodology, the students must know how to work together in teams, something the students already knew about during the first Sprint event.

In other open-ended questions, we found out what the students did in the teams, whether they got anything out of the training, how Flipboard helped them, and how much they enjoyed working with the different tools.

Figure 5 shows the answers to the questionnaire item. *How much has Flipboard helped you?* We were not surprised by the responses to this item, as we had already noticed during teaching that students were not very interested in filling out this artefact of the eduScrum methodology. The students chose their answers on a scale of 0–5, with 0 not being helpful and 5 being very helpful.

Figure 6 shows the answers to '*How much did you enjoy the following....*' The students chose their answers on a scale of 1–10, with 1 not being I didn't like it at all and 10 being super.

According to the answers, students enjoyed working in the App Inventor environment (99 students answered very much, 62 students somewhat enjoyed it, and 37 did not enjoy it). Most students (77) did not enjoy working in Android Studio, while 54 enjoyed it a little. This item confirmed our observations and pupil interviews-we need more powerful computers to teach in the Android Studio environment and more time to teach the programming language's syntax. Students enjoyed working with the micro:bit educational board (98 students said they enjoyed it a lot, 66 enjoyed it a little, and only 23 said they did not enjoy the work). The responses confirmed our assumptions and observations that

**Table 8 Hypotheses for the questionnaire item** *how would you rate your team and collaboration?*

| $H_0$ | $H_1$ | Probability of error | Value of the test criterion u | The null hypothesis |
|---|---|---|---|---|
| There are no significant differences in boys' and girls' responses on how they rate their team and cooperation. | There are significant differences in the responses of boys and girls on how they rate their team and cooperation. | 1.96 | 1.340 | Accept |
| There are no significant differences in the responses of primary and secondary school students to how they rate their teamwork and cooperation. | There are significant differences in the responses of primary and secondary school students on how they rate their teamwork and cooperation. | 1.96 | 0.350 | Accept |

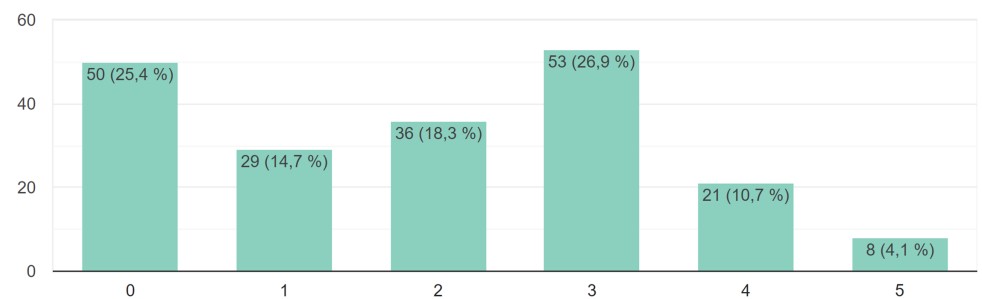

**Figure 5 The questionnaire item** *how much has flipboard helped you?*

using the micro:bit educational board excites and motivates students to learn programming. We only added game programming in MakeCode Arcade in the last academic year, so the responses are significantly fewer than for other tools. Students mostly enjoyed working in this environment-48 said they enjoyed it a lot, 30 students enjoyed it a little, and 12 students did not. This item confirmed that the students found programming games and using the micro:bit educational board motivating and exciting.

## RESULTS AND DISCUSSION

Our observations from the first year 2018/2019 showed that the students liked the stand-up event the least, which they would have skipped unless we constantly reminded them. We had problems with the computers in several secondary schools; they needed to be more powerful to use IntelliJ Idea and Android Studio. Also, for this reason, we switched to the online App Inventor environment we used in elementary school (with more difficulty).

Our observation and the questionnaire results showed that students liked the eduScrum methodology. The Android Studio and IntelliJ Idea environments were challenging for them. Because of this, we were behind schedule in all schools, and the students created elementary apps-following the tutorials on the YouTube page with minor tweaks. The teams at all schools needed the most help in the last Sprint event. The biggest problem was at the secondary school in Zvolen because 31 students participated in the lesson. In the last Sprint event of the semester, individual teams started to ask more and more questions

How much did you enjoy following:

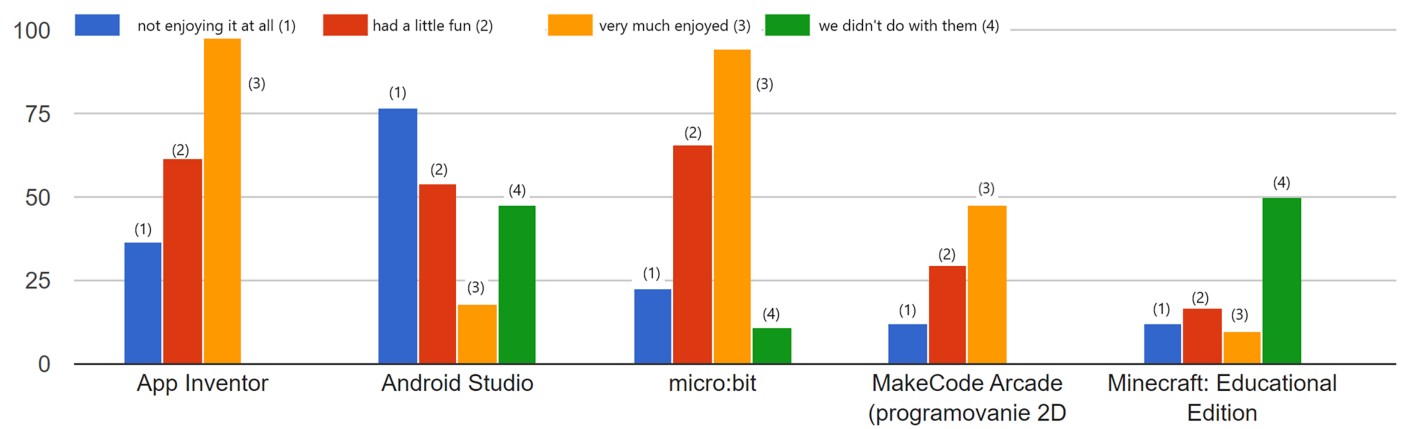

**Figure 6** The questionnaire item '*How much did you enjoy the following*?'

(there were too many students). Nevertheless, we recommend teaching the eduScrum methodology in classes with more students (unfortunately, there are very few computer science teachers in Slovakia, and it is common for a teacher to have 30 students in a class).

After our experience with the 2018/2019 academic year, we started to use App Inventor as a programming environment in secondary schools. After consulting with teachers, we chose the eduScrum teaching methodology in two secondary schools. We explained a new theory for 10–15 min in each lesson. Then, the students worked in teams on their solutions. This way, we saved time with the basic syntax of Kotlin, and the students could spend more time programming mobile apps. The rules of eduScrum can be changed if the difficulty of the curriculum requires it. We recommend explaining the basics of the subject for 10–15 min at the beginning of the lesson and then letting the students continue learning using the eduScrum methodology.

Teaching mobile applications is interesting for students. Compared to other research described in the chapter on teaching programming using mobile applications, our research confirmed that programming using the App Inventor tool is also more motivating in Slovak schools than regular programs for teaching programming. To create mobile applications, we recommend using App Inventor over other tools, such as Java, Kotlin, and Swift programming languages.

In the academic year 2020/2021, we modified the programming teaching content in some schools. We have started teaching robotics programming (micro:bit, Lego EV3, Ozobot, mBot) instead of mobile apps. Our research confirmed that these educational robots are more interesting for lower-grade elementary school students. Students enjoyed working with robots; we recommend starting work with Ozobot, Shero Bolt, and Edison robots and continuing with Lego or mBot robots.

Statistical processing of the questionnaire responses showed no significant differences between the responses of boys and girls, except for one item. The only difference was whether they thought they had improved more in programming. The statistical treatment for this one item showed us that there were differences between boys and girls, and consequently, the arithmetic mean showed that more boys than girls answered that they had improved in programming. There were no differences between the responses to the other items on the questionnaire. Statistical processing showed no significant differences between boys' and girls' responses to the items. How did you like App Inventor? Did you often need the help of a teacher? Whether they would like to continue programming applications in App Inventor. The girls liked to develop mobile apps and would like to continue programming mobile apps. These findings are interesting, as several organizations (*e.g.*, you too in IT) and teachers from regional schools reported in interviews that boys are more motivated to study computer science. The statistics of students enrolled in our Department of Applied Computer Science and Software Development show that more boys than girls study computer science. Based on the results of statistical processing, the problem of fewer girls in computer science majors may be related to the poorer motivation to study computer science in primary and secondary schools. If teachers taught programming with more innovative tools or with the help of robotic aids, girls might be more motivated to study computer science in universities.

When comparing the responses of primary and secondary school students, we found that secondary school students preferred the eduScrum methodology more than conventional teaching, more secondary school students needed help from a teacher, and more students liked the App Inventor environment compared to primary school students. Secondary school students responded that they improved and would like to continue programming mobile applications in the App Inventor environment more than secondary school students. These results were unsurprising to us, as students from secondary schools want to experiment more (responses to the item if they prefer more lessons with eduScrum methodology than regular lessons). Several students may already know what profession they want to study and are more interested in programming (responses to the item if they have improved in programming) and ask more questions (item if they often needed help from a teacher).

The primary purpose of teaching using the eduScrum methodology is the development of soft skills. In several questionnaire items, we asked about the development of soft skills:

- How would you rate your team and cooperation?
- Are you good to work with? Are you a team player?
- Did you do anything creative in class? If yes, what was it?
- Can you be more creative in an eduScrum class than a regular one?
- What did these lessons give you? Fun, personal growth, motivation for computer science, collaboration
- Is planning important in eduScrum?

These questions showed that most of the students were doing something creative, were optimistic about themselves and the cooperation of other team members, and that planning was essential. From the answers, we can conclude that the eduScrum methodology improves the so-called soft skills of the students. We also saw an improvement in these skills in the goal presentations, where, in the beginning, the students' presentations were less well prepared, poorly timed, and less well presented than the goal presentations in the last Sprint events.

Compared with similar research described at the beginning of the article, our research confirmed that eduScrum teaching (if it is well-prepared by the teacher) is more interesting for students than regular frontal teaching. The disadvantage of this methodology is the necessity of preparing better teaching materials (and more preparation of the teacher before the lesson). The teacher must be an expert in the given topic, as the teams go at different paces; some students want to create more difficult software (applications, the behaviour of robots) than if they were taught through traditional teaching (he must be able to help).

Academic performance of students using the eduScrum methodology were the same compared to previous years. This is because students usually have the best grades in Slovak schools.

## CONCLUSIONS

Primary and secondary schools in Slovakia teach mainly in the traditional way-frontal teaching. For a long time, various analyses, research, and comparisons have mentioned the necessity to change the form of education in Slovak primary and secondary schools. The main change should be in shifting the focus of education from the transfer of knowledge to the development of students' competencies, adapting school educational programs to individual possibilities and current social challenges and needs. This article describes the eduScrum methodology based on the most popular software development method, Scrum. We described the principles and our experience with this methodology. EduScrum strengthens students' soft skills and could help improve students' motivation toward the subject. We have tested the suitability of the eduScrum methodology over three academic years in some primary and secondary schools in the Banská Bystrica area. During the validation, we investigated whether there were significant differences after the completion of the teaching with this methodology between the responses of boys and girls and between the responses of students from primary and secondary schools. Based on the statistical evaluation of our research, we can conclude that there are no significant differences between boys' and girls' responses to the eduScrum methodology; the only difference was that boys responded that they had improved more in programming than girls. When comparing the responses of primary and secondary school students, it was found that secondary school students preferred the eduScrum methodology more than traditional frontal teaching, needed more help from the teacher, and liked developing mobile applications in the App Inventor environment more. In the future, we plan to continue teaching the eduScrum methodology at primary and secondary schools to compare them with the teaching of the methodology at universities.

### Funding

This contribution has been processed as part of grant project no. 001UMB-4/2023 Implementation of blended learning in the training of professional bachelors and teachers of mathematics and computer science. The funders had no role in study design, data collection and analysis, decision to publish, or preparation of the manuscript.

### Grant Disclosures

The following grant information was disclosed by the authors:
Grant Project: 001UMB-4/2023.

### Competing Interests

The author declare that they have no competing interests.

### Author Contributions

- Patrik Voštinár conceived and designed the experiments, performed the experiments, analyzed the data, performed the computation work, prepared figures and/or tables, authored or reviewed drafts of the article, and approved the final draft.

### Data Availability

The raw data is available in the Supplemental File.

### Supplemental Information

Supplemental information for this article can be found online at http://dx.doi.org/10.7717/peerj-cs.1822#supplemental-information.

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
