# Peer review of "Teaching programming using eduScrum methodology"

_PeerJ Computer Science, doi:10.7717/peerj-cs.1822_

## Round 0.1 · original submission · Minor Revisions

Please consider the comments made by the two reviewers in order to prepare a new version of the manuscript.

Reviewer 1 ·

Basic reporting

Overall, the document is easy to follow. I have only a few minor suggestions:

1. The authors could include some ideas/main conclusions from the analysis of results in the last part of the introduction.

2. Until the discussion I did not fully understand that some students worked with the mobile development environment while others applied the eduScrum methodology to robotic kits. The authors could indicate this aspect as part of the methodology section.

Experimental design

1. It would be interesting to know whether the results are independent of the course subject. The authors could analyse the results by dividing the students into those who applied eduScrum to mobile app development and those who applied eduScrum to robotics kits.

2. The author should specify the sample of students belonging to each population group used to analyse the results, i.e., how many boys/girls and how many primary/secondary students responded to the questionnaire.

3. It is not clear whether all questions in the questionnaire are presented as a yes/no question, or the authors considered a likert scale (low disagreement - high disagreement). I found this information in the questionnaire provided as additional material. Some questions have 5 options while others have 4 options or are simply yes/no questions. The author should indicate how the positive/neutral/negative values are aggregated to calculate the statistical tests.

Validity of the findings

The author could enrich the experience report included in the discussion section with some evidence of the academic performance of students using the eduScrum methodology. It would be interesting to know whether, apart from a better student experience, the use of the eduScrum methodology had a positive impact on the grades obtained by students compared to previous years.

Additional comments

The author includes valuable supplementary material. If the material will be available after acceptance, I recommend adding a few lines referring to the availability of such material in the methodological section for the reader's reference.

Cite this review as
Anonymous Reviewer (2024) Peer Review #1 of "Teaching programming using eduScrum methodology (v0.1)". PeerJ Computer Science

Reviewer 2 ·

Basic reporting

The paper presented is well-written, and even if you're not a native English speaker, you can understand the text, its objectives, description and conclusions.
The abstract summarizes the work, but should end with a conclusion on the results obtained.
The paper presents a good framework for the topic, describing examples of other tools and technologies that can be or are used in the context of early programming learning.
Figure 1: The small menu at the top of the image doesn't make sense. I think the aim is to show how easy it is to use. There should therefore be a brief description on the image highlighting these features.
In figure 2: There should be identification of the different robots.
Tables are usually identified at the top of the table, not in the footer like the figures.
The content of table 1 could be more user-friendly... The font size is too large and difficult to read.

Experimental design

For the defined objectives: "The research aims to determine if teaching computer science in primary and secondary schools using eduScrum is more effective than classical-frontal teaching." The work presents satisfactory results and evaluation.
The method described presents a satisfactory description.

Validity of the findings

The idea of using eduScrum is an interesting one, and it attracts the attention of all those who deal with the problem of difficulties in teaching and learning programming. However, it seems to me that this type of approach for elementary school pupils is not the most appropriate, since these pupils are not mature enough to develop this type of Srum development.
With the minor corrections pointed out, I am in favor of its publication. It will be a good contribution to development in the area.

Cite this review as
Anonymous Reviewer (2024) Peer Review #2 of "Teaching programming using eduScrum methodology (v0.1)". PeerJ Computer Science

---

## Round 0.2 · accepted · Accept

Congratulations on a good job. Both reviewers considered this new version is perfect.

Reviewer 1 ·

Basic reporting

No additional comments.

Experimental design

No additional comments.

Validity of the findings

No additional comments.

Additional comments

The author has addressed my previous comments, so the paper is ready for publication.

Cite this review as
Anonymous Reviewer (2024) Peer Review #1 of "Teaching programming using eduScrum methodology (v0.2)". PeerJ Computer Science

Reviewer 2 ·

Basic reporting

no comment

Experimental design

no comment

Validity of the findings

no comment

Additional comments

no comment

Cite this review as
Anonymous Reviewer (2024) Peer Review #2 of "Teaching programming using eduScrum methodology (v0.2)". PeerJ Computer Science